# Engineering Herbicide-Tolerance Rice Expressing an Acetohydroxyacid Synthase with a Single Amino Acid Deletion

**DOI:** 10.3390/ijms21041265

**Published:** 2020-02-13

**Authors:** Jun Fang, Changzhao Wan, Wei Wang, Liuyin Ma, Xinqi Wang, Can Cheng, Jihua Zhou, Yongjin Qiao, Xiao Wang

**Affiliations:** 1Crop Breeding and Cultivating Institute, Shanghai Academy of Agricultural Sciences, 1000 Jingqi Rd, Shanghai 201403, China; wanchangzhao@163.com (C.W.); zw10@saas.sh.cn (X.W.); chengcan222@126.com (C.C.); zhoujihua@saas.sh.cn (J.Z.); nkseed2010@aliyun.com (Y.Q.);; 2Shanghai Institute for Advanced Immunochemical Studies, ShanghaiTech University, 99 Haike Rd, Shanghai 201210, China; kimi-ww@126.com; 3Basic Forestry and Proteomics Research Center, Haixia Institute of Science and Technology, Fujian Agriculture and Forestry University, Fuzhou 350002, China; lma223@fafu.edu.cn

**Keywords:** AHAS activity, the W548 deletion, multi-herbicide tolerance, rice

## Abstract

The acetohydroxyacid synthase (AHAS) is an essential enzyme involved in branched amino acids. Several herbicides wither weeds via inhibiting AHAS activity, and the *AHAS* mutants show tolerance to these herbicides. However, most AHAS mutations are residue substitutions but not residue deletion. Here, residue deletion was used to engineering the *AHAS* gene and herbicide-tolerant rice. Molecular docking analysis predicted that the W548 of the AHAS was a residue deletion to generate herbicide tolerance. The AHAS-ΔW548 protein was generated in vitro to remove the W548 residue. Interestingly, the deletion led to the tetramer dissociation of the AHAS, while this dissociation did not reduce the activity of the AHAS. Moreover, the W548 deletion contributed to multi-family herbicides tolerance. Specially, it conferred more tolerance to sulfometuron-methyl and bispyribac-sodium than the W548L substitution. Further analysis revealed that AHAS-ΔW548 had the best performance on the sulfometuron-methyl tolerance compared to the wild-type control. Over-expression of the *AHAS-ΔW548* gene into rice led to the tolerance of multiple herbicides in the transgenic line. The T-DNA insertion and the herbicide treatment did not affect the agronomic traits and yields, while more branched-chain amino acids were detected in transgenic rice seeds. Residue deletion of W548 in the AHAS could be a useful strategy for engineering herbicide tolerant rice. The increase of branched-chain amino acids might improve the umami tastes of the rice.

## 1. Introduction

Herbicide tolerance is an important trait of biotech crops worldwide [1]. Over the past 20 years, the strategy combining herbicide-tolerant (HT) crops with specific herbicides has made a significant contribution to weed control [2,3]. This system is reliable and desired by the market [4]. In the paddy, this strategy was the only economic and effective method to control weedy rice. Weedy rice has similar taxonomic and physiological features with rice. Some weedy rice is de-domesticated from commercial rice [5]. Since rice transplanting cultivation was shifted to direct sowing in Asia [6], weedy rice has become an important problem [7]. Penoxsulam, cyhalofop-butyl and pretilachlor are the most used herbicides in paddy [8]. Penoxsulam inhibits acetohydroxyacid synthase (AHAS, EC2.2.1.6, also known as acetolactate synthase). It eliminates weeds but not rice because rice Cytochrome P450 oxidase catalyzes O-dealkylation to 5-OH-penoxsulam and leads to herbicide degradation [9]. Cyhalofop-butyl is an inhibitor of acetyl coenzyme-A carboxylase (ACCase) but it can be metabolized and oxidized to form nontoxic diacid in rice [10]. Pretilachlor inhibits germination of plants including rice, and it is only used as a pre-planting or post-emergence application [11]. However, those herbicides are specialized for rice and are detoxified in rice. They are not effective for weedy rice [2]. The HT rice is engineered to tolerate the herbicides which could kill non-biotech rice cultivars. Imidazolinone-tolerant rice displays a promised strategy to control weedy rice in paddy fields [12]. Besides imidazolinone (IMI), other herbicide families, such as sulfonylurea (SU) and pyrimidinyl-benzoate (PYB), are popular in global agriculture [13]. The target of those herbicides is AHAS. Those herbicides control the grass and broadleaf weeds including weedy rice and are ideal candidates for the development of HT rice [2].

The AHAS mutants have been used to develop herbicide-tolerant crops. The AHAS is the first enzyme in the biosynthesis of three branched-chain amino acids (BCAA): valine, leucine, and isoleucine [14]. This enzyme is essential in plants and microorganisms but is absent in animals. Thus, the AHAS-inhibiting herbicides are non-toxin to animals. The herbicides bind to AHAS, inhibit catalytic efficiency and decrease BCAA contents [15]. This process retards plant growth or kills plants [16]. However, a weed evolves tolerance to SU due to a proline to histidine mutation in AHAS [17,18]. The mutation reduces the binding affinity between AHAS and SU, which produced herbicide tolerance. Learning from nature, scientists introduce various *ahas* mutants to develop HT crops [19]. Induced mutagenesis has been used to develop tolerant crops since 1992 [12]. The different mutations generate SU, IMI, or PYB tolerance in maize, sunflower, rice, wheat, and oilseed rape. However, the IMI-tolerant rice has been applied for 18 years, the weedy rice evolves IMI tolerance in Italy [20]. Novel HT rice is needed to deal with this problem.

The HT rice was engineered to tolerate multi-family herbicides through a residue deletion in the AHAS. The deletion was uncommon mutations because it led to protein degradation in certain cases [21]. Previous mutations in AHAS were substitution but not deletion [16]. The W548 residue (in this study, the amino acid numbering is based on rice AHAS) was an important site to generate herbicide tolerance in AHAS [14]. Its substitutions had been reported in many organisms, such as plants, bacteria, and yeasts. But it was unclear whether the W548 deletion led to herbicide tolerance in AHAS. Molecular docking is a method to predict the orientation and location of a small compound in a protein [22]. An algorithm was conducted to evaluate a series of compound-protein complexes to obtain the one with minimum energy. The complex could display the surface of the binding site and the conformation of the compound. We docked several herbicides in rice AHAS to study the interactions between the W548 and those herbicides. The W548 was removed in rice AHAS, then this modified enzyme (AHAS-ΔW548) were characterized in vitro. Transgenic rice was developed to evaluate the effects of *ahas-ΔW548* gene on plant traits.

## 2. Results

### 2.1. Herbicide Tolerance Predicted in AHAS Models

Stereo models of the wild-type AHAS (AHAS-WT, NCBI, GenBank ID: BAB20812) were built with SU, IMI, and PYB herbicides. The SU family included four herbicides: sulfometuron-methyl (SM), rimsulfuron (RS), chlorimuron-methyl (CM), and flucarbazone-sodium (FC). The PYB and IMI families included bispyribac-sodium (BS) and imazethapyr (IT), respectively. In the AHAS-WT, the indole ring in the W548 faced with the triazine (FC) or the pyrimidine (SM, RS, CM, and BS) ring (Figure 1). Those face to face rings could form the π-π interaction, which anchored the herbicides in the AHAS-WT. The W548 was far away from the IT which bound to the protein with S627 [23]. The herbicides blocked the channels which substrates passed into the catalytic centers in the AHAS-WT. After deleting the W548, the scores dropped more than 10% for five herbicides (Table 1). Due to a lack of homologous crystal structures of the AHAS-ΔW548, molecular docking could not produce precious structures. The scores implied that the W548 deletion might weaken the interaction and change the channel conformation. Although no interaction was found between the W548 and the IT, the W548 deletion opened the mouth of the channel. Those results supposed that the W548 deletion could lead to herbicide tolerance. To verify this prediction, the AHAS-ΔW548 was expressed and characterized in vitro to examine the effects of the deletion.

### 2.2. The W548 Deletion Dissociate the Tetramer in Vitro

The AHAS-GST and mature AHAS proteins in a gel showed high purity of proteins (Figure 2). The GST-AHAS and tag-free AHAS were observed at predicted molecular weights of 90 kDa and 64 kDa, respectively. Gel filtration chromatography revealed clear peaks of expected molecular weight (Figure 3). No aggregation or degradation peaks were observed. Those results indicated that purified proteins were homogeneous and well folded. The AHAS-WT trace consisted of two peaks: one main dimer peak (128 kDa) and one small tetramer peak (256 kDa). Only one dimer peak was observed in the AHAS-ΔW548 trace. The tetrameric peak was not detected for the AHAS-ΔW548. This suggested that the residue deletion led to the dissociation of a tetramer into dimers.

### 2.3. Multi-Herbicide Tolerance of the AHAS-ΔW548

The activity of the AHAS-WT decreased as herbicide concentration increased (Figure 4). Obviously, the AHAS-ΔW548 remained activities at maximum concentrations of herbicides. Especially, the mutated AHAS displayed high activities in the SU solutions. Moreover, the AHAS-ΔW548 conferred good tolerance to the PYB/IMI. Kinetic parameters showed no significant variation for the AHAS-ΔW548 in the absence of an herbicide (Table 2). Similar results were observed in the presence of three SU herbicides (SM/RS/CM). The catalytic efficiency of the AHAS-ΔW548 was inhibited by FC, BS, and IT. The reaction velocity (*K_cat_*) of the AHAS-ΔW548 decreased significantly in the FC solution, suggesting that FC was bound and blocked the channel in the mutant. The BS or IT significantly inhibited the reaction velocity and binding affinity of the mutated protein. It implied that the BS or IT bound to the channel and affected the active site. Among all herbicides, SM inhibited the AHAS-WT at a low concentration, but did not affect the AHAS-ΔW548 at a high concentration. Thus, SM was a good reagent to select the tissues expressing the *ahas-ΔW548* gene.

### 2.4. Transgenic Rice with Herbicide Tolerance

The T-DNA cassette expressing the *ahas-ΔW548* gene (Figure 5) was combined with SM to select HT callus and shoots in tissue culture procedure. Twenty-one lines are positive in the T0 generation after the polymerase chain reaction (PCR) scan (Figure 6). The T1 plants from eighteen lines survived with the SM treatment. The levels of the *ahas* expression indicated that the *ahas-ΔW548* gene was over expressed in those survived lines (Appendix A). Some lines displayed accepted agronomic traits (Appendix A), but the Line9 was the best one for comprehensive traits. The T2 plants of Line9 did not display significantly difference with its parent Xiushui134 in field (Table 3). Moreover, the Line9 showed tolerance to multi-family herbicides (Figure 7). Relative heights were not affected by all SU herbicides (Figure 8). Line9 was dwarfed with the BS or IT treatment. The SM treatment had no effect on the agronomic traits of the Line9. Contents of total proteins in brown rice were similar between the Xiushui134 and the Line9 (Table 4). Contents of BCAA increased significantly in the Line9, while other free amino acids kept similar contents. The SM treatment did not change the BCAA contents in brown rice.

## 3. Discussion

### 3.1. Effects of the W548 Deletion on Rice AHAS

Tetramer dissociation was an unexpected feature of the AHAS-ΔW548. The S627 position was close to the W548 in the structure; they located distant from the tetramer interface (Figure 9). However, the S627N mutation did not lead to tetramer dissociation [23]. The W548 deletion was not predicted to break the interface. The W548 deletion might lead to structure rearrangement and indirectly disrupt the tetramer interface. Catalytic efficiency depended on the active site in a dimer [24]. The remained catalytic efficiency indicated that the AHAS-ΔW548 dimer contained an intact active site. The W548L mutant was used widely in HT crops [14]. When a herbicide’s concentration was 100 μM, the W548L mutant remained about 20% activity in the SM solution and less than 10% activity in the BS solution [25,26]. At this concentration, the AHAS-ΔW548 remained about 100% activity in the SM solution and more than 60% activity in the BS solution. It implied that the W548 deletion had more tolerance to SM and BS than the W548L substitution. The enhanced tolerance might be a result of the gap at the mouth of the AHAS-ΔW548 channel. Meanwhile, no gap happened in the W548L substitution.

### 3.2. Mechanism of Herbicide Tolerance for the AHAS-ΔW548

Herbicide tolerance was generated when some mutations weakened the binding between AHAS and an herbicide [14]. A SU/PYB herbicide was bound to the W548 with a π-π interaction [27,28]. When the W548 was replaced with a non-aromatic residue, the π-π interaction disappeared and herbicide tolerance was generated [29,30]. Herbicide tolerance also occurred when the W548 was replaced with other aromatic residues [31]. The benzene ring in tyrosine or phenylalanine changed the orientation and dihedral angle, reducing the π-π interaction. The AHAS channel varied its conformation to the herbicide binding [28]. The loss of the π-π interaction of the AHAS-ΔW548 was supposed to change the protein’s conformation. This variation might shift the herbicide position and allow substrates access to the active center. The BS induced the cleavage of a thiazolium ring, which reduced the cofactor content and binding affinity [15]. It was the largest molecule in all tested herbicides and occupied more space in the channel. This occupation could cause the reduction of the reaction velocity of the AHAS-ΔW548. The binding between IT and AHAS caused a negative effect on the active site [28]. This effect might decrease binding affinity and the passing speed of substrates.

### 3.3. Development of HT Genes Using Residue Deletion

Besides the W548 mutations, several mutations in AHAS conferred tolerance to herbicides [14]. The S627N mutation conferred tolerance to IMI [23]. The AHAS mutations, such as K256F, M351C, H352Q, and F578D, led to tolerance to SU, IMI, and triazolopyrimidine [16]. Other endogenous enzymes were targeted by the herbicides which action modes were different from AHAS-inhibiting herbicides. Those enzymes included photosystem two complex, ACCase, 5-enolpyruvylshikimate-3-phosphate synthase (EPSPS), tubulin, and protoporphyrinogen IX oxidase [32]. Amino acid substitution in those enzymes led to herbicide tolerance. Those mutations could be candidates of single amino acid deletion. The deletion probably conferred herbicide tolerance to an enzyme. The W548L/S627I double mutation performed high BS tolerance of rice AHAS [33]. Moreover, simulated structures and herbicide docking could check the possibility of herbicide tolerance in modified enzymes. The deletion of two or more residues would be engineered to pursue extra herbicide tolerance in the further research.

### 3.4. The Effects of the Ahas-ΔW548 Gene on the Line9

Transgene position affected parent phenotypes when T-DNA inserted into a certain functional gene [34]. No change of the phenotype demonstrated that the T-DNA position in the Line9 genome did not produce negative effects. Overexpression of some endogenous gene reduced rice yields [35]. The similar traits between Line9 and its parent suggested that AHAS-ΔW548 overexpression did not reduce rice yield in this line. The Line9 height with FC treatment suggested that the residual activity of the AHAS-ΔW548 is sufficient to maintain rice normal growth. A P450 monooxygenase in wild-type rice could degrade the BS to some degree [36], thus the plants with the BS treatment were higher than those with the IT treatment. Weedy rice had tolerance to the herbicides which were detoxified by non-biotech rice [2]. It was eliminated by the herbicides which killed non-biotech rice. Death of Xiushui134 implied that the tested herbicides could eliminate weedy rice. The SM treatment in the fields indicated that this herbicide could be used to control weeds and produce accepted yields. A substitution mutant enhanced BCAA contents to be two-fold in the seeds [37]. The AHAS-ΔW548 overexpression in Line9 displayed more fold than the substitution mutant. The BCAA compounds were of bitter taste [38]. However, the bitter thresholds of the valine, leucine, and isoleucine were 3.4 mM, 11 mM, and 10mM, separately. A bitter amino acid enhanced the umami taste of foods at subthreshold concentrations [39]. The BACC contents of Line9 seeds were lower than their bitter threshold. It is a possibility that those BACC could enhance the umami taste of the rice.

### 3.5. Novel Herbicide-Tolerant Biotech Crops

Novel HT crops are desired in the market [1]. The EPSPS has been utilized as the target of glyphosate for dozens of years [40]. Weeds evolved glyphosate tolerance in soybean fields. Transgenic soybean expressing an IMI-tolerant AHAS was developed to control those weeds [41]. Current HT rice had high tolerance to IMI but not to SU [12]. The weedy rice evolving IMI tolerance could be controlled by the SU-tolerant Line9. Recently, precise gene modification efficiently generated HT crops. The HT rice was engineered through introducing mutations into endogenous *ahas* gene using the transcription activator-like effector nucleases (TALEN) [33] and the clustered regularly interspaced short palindromic repeats (CRISPR) mediated mutagenesis [42]. CRISPR technology introduced a mutation into endogenous *EPSPS* gene and produced glyphosate-tolerant rice [43]. Crops could develop herbicide tolerance by deleting single amino acid from an endogenous gene. In the future, the W548 codon would be removed from rice endogenous *ahas* gene to generate herbicide tolerance.

## 4. Materials and Methods

### 4.1. Structural Simulation of Rice AHAS-Herbicide Complexes

Structures of rice AHAS were predicted using the online server SWISS-MODEL [44]. The AHAS sequence of japonica rice was found in the National Center for Biotechnology Information (NCBI, GenBank ID: BAB20812). Although no structure of rice AHAS has been determined, crystallographic structures of Arabidopsis AHAS are deposited in the Protein Data Bank (PDB). Structures of rice AHAS are simulated using homologous complexes of Arabidopsis. Herbicide molecules were retrieved from the PubChem database (Appendix A). They were docked into the corresponding structures using the Maestro (v10.2) software bundle (Schrödinger L.L.C., New York, NY). The Glide program was used to predict the binding conformation in herbicide-AHAS complexes [45].

### 4.2. Purification of the AHAS Proteins

General experiments were similar as previously described with minor modifications [23]. The *ahas* sequence of japonica rice was found in the NCBI (GenBank ID: AB049822). The *ahas-WT* was amplified from japonica rice (Xiushui134) using the PCR. The primers were AHAS-F and AHAS-R (all sequences of primers were synthesized by Shanghai Sangon Biotech and were listed in Appendix A). Site-directed mutagenesis kit (Qiagen, Lane Valencia, CA, USA) was used to delete the W548 codon and generate the *ahas-ΔW548*. The primers were AHAS-W548F and AHAS-W548R. The sequences of *ahas-WT* and *ahas-ΔW548* were verified through DNA sequencing and the Vector NTI software (ThermoFisher Scientific, Waltham, MA, USA). Signal peptides of the N-terminal 59 amino acids were removed by the PCR to express the mature AHAS in *Escherichia coli* [46]. The primers were AHAS-BamHI-F and AHAS-EcoRI-R. The *ahas* genes were cloned into pGEX4T2 (GE Healthcare, Pittsburgh, PA). The resultant plasmids were transformed into *E. coli* Rosetta (Novagen, Sacramento, CA, USA) cells for protein expression. When cell culture grew to the log phase, protein expression was induced with 1 mM isopropyl β-D-1-thiogalactopyranoside (chemicals in this study were purchased from Sigma-Aldrich Corp., St. Louis, MO, USA) and incubated with shaking at 25 °C for 6 h. The AHAS protein fused with an N-terminal glutathione S-transferase (GST) tag was isolated using a glutathione agarose column. The fusion protein was cleaved by a Thrombin CleanCleave Kit (Sigma-Aldrich Corp.). The resultant mixture was passed through the same column again. Mature AHAS proteins were isolated in the flow-through fraction and were analyzed using sodium dodecyl sulfate-polyacrylamide gel electrophoresis (SDS-PAGE). Gel filtration chromatography was used to determine oligomeric states and examine the homogeneity of purified proteins [47]. A high-performance liquid chromatography system and an Ultrahydrogel 1000 SEC column (Waters Corp., Milford, Massachusetts) were used in the experiment.

### 4.3. Activities and Kinetic Assays of Rice AHAS in Vitro

Activity and kinetic assays were conducted following a reported method [48]. An assay buffer contained 50 mM phosphate buffer (pH 7.5), 10 mM MgCl_2_, 20 μM flavin adenine dinucleotide, 1 mM thiamine pyrophosphate, 40 mM pyruvate, and 1 μM purified AHAS. Herbicides were added into the reaction at specific concentrations. After incubation at 37 °C for one hour, H_2_SO_4_ was added to a final concentration of 1% to stop the reaction. Produced acetolactate was decarboxylated to generate acetoin at 60 °C for 30 min. The acetoin was incubated with 0.5% creatine and 5% α-naphthol at 60 °C for 15 min. The mixture changed color from yellow to red. Absorbance was detected at 525 nm (A_525 nm_). The A_525 nm_ value (A_Bk_) without the enzyme was considered 0%. The A_525 nm_ value (A_WT_) for AHAS-WT was recorded in the absence of an herbicide. The value (A_100_) of A_WT_ minus A_BK_ was considered 100%. The A_525nm_ value (A_S_) of a sample was recorded. The percentage of remaining AHAS activity (R_%_) was calculated by the Equation (1): (1)$$R_{\%}\;=\;\left(A_{S}-A_{\mathrm{Bk}}\right)\times 100/A_{100}$$

Continuous assays explored the effects of the W548 deletion on AHAS catalytic efficiencies. Kinetic values were determined by differing pyruvate concentrations. An herbicide was added at a concentration of 100 μM. The assay measured the reaction velocity (*v*) at different the pyruvate concentration ([*S*]) with the fixed concentration of the AHAS dimers ([*E*]). The Michaelis–Menten constant (*K_m_*) and catalytic rate constant (*K_cat_*) were obtained through the Equation (2): (2)$$v\;=\;V_{max}\;\times \;\left[S\right]/\left(K_{m}\;+\;\left[S\right]\right)\;\mathrm{and}\;K_{cat}\;=\;V_{max}/\left[E\right]$$

The catalytic efficiency was calculated by dividing *K_cat_* by *K_m_*. The activity or kinetic assay was conducted in triplicate.

### 4.4. Development of Transgenic Rice

The *ahas*-*ΔW548* gene was used as a selection marker in the transformation procedure. A T-DNA cassette based on the vector pCAMBIA1300 (1300 for short, Cambia, Australia) was assembled to express the AHAS-ΔW548 in rice. The hygromycin-resistant gene was removed from the 1300 vector with XhoI. The digested sites were dephosphated with alkaline phosphatase. The modified vector was named 1300-XhoI-DP. The *ahas-ΔW548* was amplified with the primers AHAS-XhoI-F and AHAS-XhoI-R. The PCR product was digested with XhoI and cloned into the 1300-XhoI-DP. The correct direction of the *ahas-ΔW548* was confirmed with DNA sequencing. The resultant construct was named 1300-AHAS-ΔW548 and was transformed into an *Agrobacterium tumefaciens* stain (LBA4404, ThermoFisher Scientific). Xiushui134 calluses were used as a recipient. Rice transformation was conducted following a previous protoco l [49]. The selection agent was 1 mM SM, applied in the process of tissue culture. Transgenic plants were scanned with the PCR in the T0 generation. The genomic DNA of Xiushui134 was used as a negative control. Plasmid DNA of the 1300-AHAS-ΔW548 was used as a positive control. The primers AHAS-1665F and AHAS-1717R were designed to distinguish the *ahas-ΔW548* (50bp) fragment from *ahas-WT* (53bp). The PCR fragments were resolved on a 20% PAGE gel. Seeds were harvested from positive plants.

### 4.5. Herbicide Tolerance and Agronomic Traits of Transgenic Lines

The T1 plants were sprayed with 0.25 g/L SM at the seedling stage. Survival lines were considered HT rice. The relative expression of those plants was measured at the maturity stage by Quant Studio6 Real-time PCR system (ThermoFisher Scientific) [23]. Trizol solution was used to extract total RNA from rice leaf. M-MuLV First Strand cDNA Synthesis Kit (Shenggong Inc. Shanghai) was used to transcribe RNA to cDNA. Primers AHAS-341F/AHAS-500R and primers actin-F/actin-R were used to amplify the target and reference, separately. Level of *ahas* mRNA expression in the Xiushui134 was treated as 1.0. Expression were presented as the relative mRNA level. Every sample of a plant was analyzed in triplicate. Samples from five plants were detected for each line. Agronomic traits were observed in fields. The T1 seeds were harvested from the line for which the traits were similar to Xiushui134. The T2 plants were treated with 0.25 g/L SM, 0.17 g/L RS, 0.20 g/L CM, 0.09 g/L FC, 0.8 g/L BS or 2.2 g/L IT at the seedling stage in a greenhouse. In an herbicide treatment, four plants grew in a plate and ten plants were treated with an herbicide. Xiushui134 without the herbicide treatment was used as a positive control. Two weeks later, the value (H) of the plant height was recorded. The height value (H_100_) of positive control was considered 100%. The percentage (H_%_) of the relative height was calculated by the Equation (3): (3)$$H_{\%\;}=\;H\;\times \;100/H_{100}$$

Additionally, the T2 plants were characterized in weed-free fields. In one field, the transgenic line was treated with 0.25 g/L SM at the seedling stage. In other fields, the transgenic line and Xiushui134 were grown without the herbicide treatment [50]. Five plots of 1.0 m^2^ were chosen randomly for traits evaluation in one field. Every plot contained 30 plants. We measured the plant height and panicle numbers per plant at the maturity stage in fields. Panicles of an individual plant were collected to determine the seed-setting rate, grains per panicle and yield per plant in a laboratory. A QM3 rice analyzer system (Vibe, Bnei Brak, Israel) was used to measure 1000-grain weight, grain width, and grain length.

### 4.6. Measurement of Total Protein and Free Amino Acids in Rice Seeds

One kilogram of the Line9 seeds was weighted randomly from the seeds harvested from fields. They were ground into powder by a grinder (Dickey John, Auburn, IL). Total proteins were calculated from the nitrogen content: protein% = N%/0.16. The nitrogen content was measured with the Kjeldahl method [51]. The analysis was applied according to the manual of the Kjeltec 8400 instrument (FOSS, Hovedstaden, Danmark). To determinate the free amino acid content, the rice powder was added into 50% (*v*/*v*) methanol containing piperazine-1,4-bisethanesulfonic acid and methionine sulfone [37]. The mixture was centrifuged at 13,000× g for 10 min, the supernatant passed through a 5 kDa Amicon Ultra centrifugal filter unit (Merck, Darmstadt, Germany). The solution was loaded into an automatic amino acid analyzer (model L-8900; Hitachi, Tokyo, Japan) with cation-exchange chromatography. Amino acid standards were set up to calculate the contents. Each sample of the powder was repeated in triplicate. Three samples were analyzed in each experiment.

### 4.7. Data Analysis

The kinetic parameters of the AHAS-ΔW548 were compared with those of the AHAS-WT using the student’s *t*-test [23]. The agronomic traits and free amino acid content were compared with those of Xiushui134 using the same test. The bioassay curves and plant height columns were generated by the Excel of Office 2016.

## 5. Conclusions

The Line9 plants expressing the AHAS-ΔW548 showed good agronomic traits with the herbicide treatment. Although transgenic rice was limited in laboratories by the Chinese government, the transgene-free CRISPR technology could directly modify a gene in a genome and produced crops without any transgene. The Line9 proved that the *ahas-ΔW548* gene was a good candidate for endogenous modification to develop non-transgenic rice with herbicide tolerance.

## Figures and Tables

**Figure 1 ijms-21-01265-f001:**
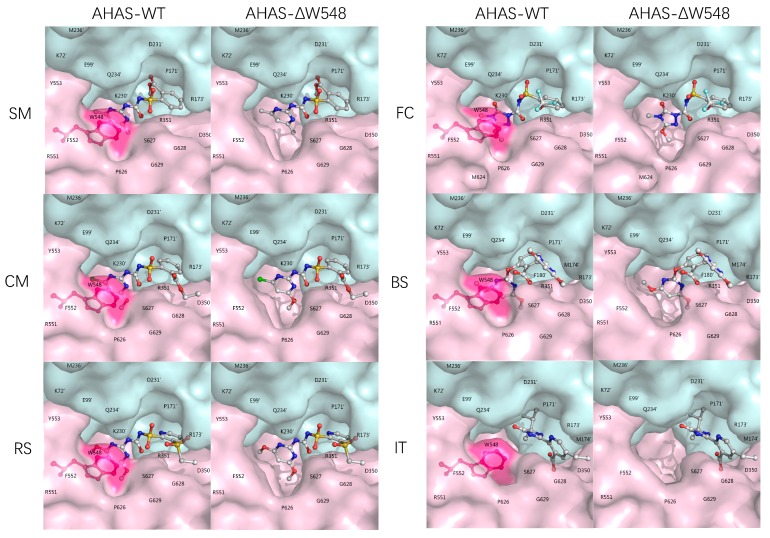
Herbicides bind and block the channel leading to the active site. The molecular surfaces of the monomers were depicted as pink and cyan, respectively. The residues were labeled on the surfaces. ’ indicated residues from different monomers. W548 was shown as a red stick-ball model with the red surface. The herbicides were shown as color stick-ball models with white carbon atoms, blue nitrogen, red oxygen, cyan fluorine, yellow sulfur, and green chloride.

**Figure 2 ijms-21-01265-f002:**
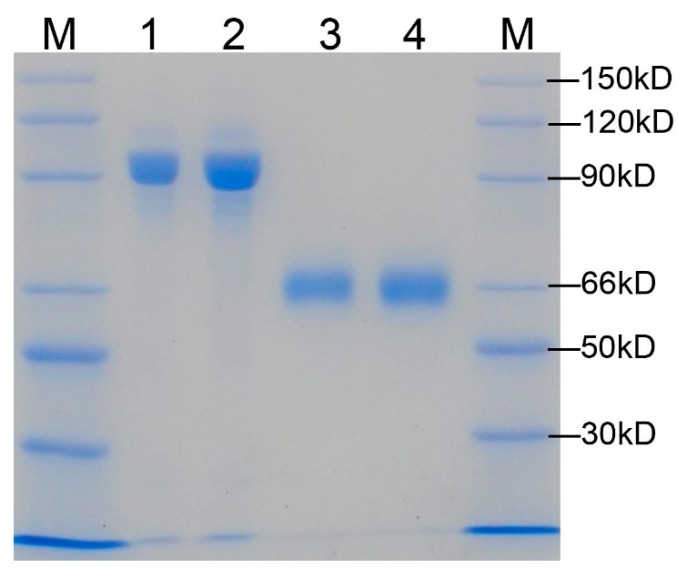
Gel image of SDS-PAGE for the purified AHAS. Lane 1: GST-AHAS-WT; Lane 2: GST-AHAS-ΔW548; Lane 3: mature AHAS-WT; Lane 4: mature AHAS-ΔW548. M: protein marker. The molecular weights of bands in the marker were indicated.

**Figure 3 ijms-21-01265-f003:**
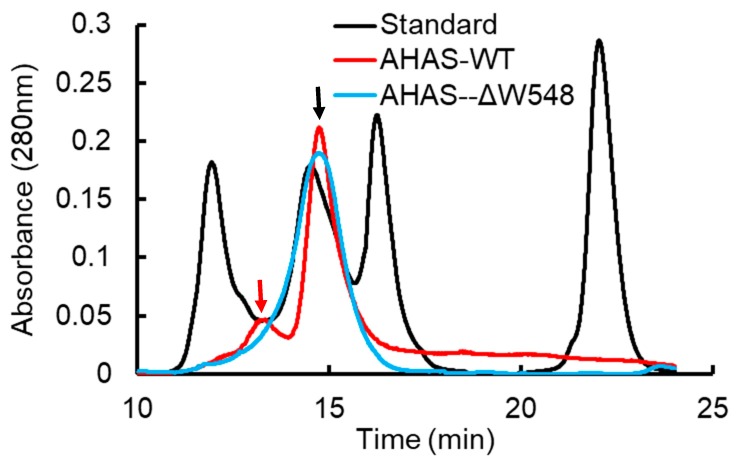
Gel filtration chromatography analysis of purified AHAS. AHAS-WT and AHAS-ΔW548 are shown as red and blue traces, respectively. The black trace is the molecular weight standard. From left to right, the molecular weights of the standards were 670, 158, 44 and 1.35 kD. The dimer peak (128 kDa) and tetramer peak (256 kDa) are indicated separately by the black arrow and red arrow.

**Figure 4 ijms-21-01265-f004:**
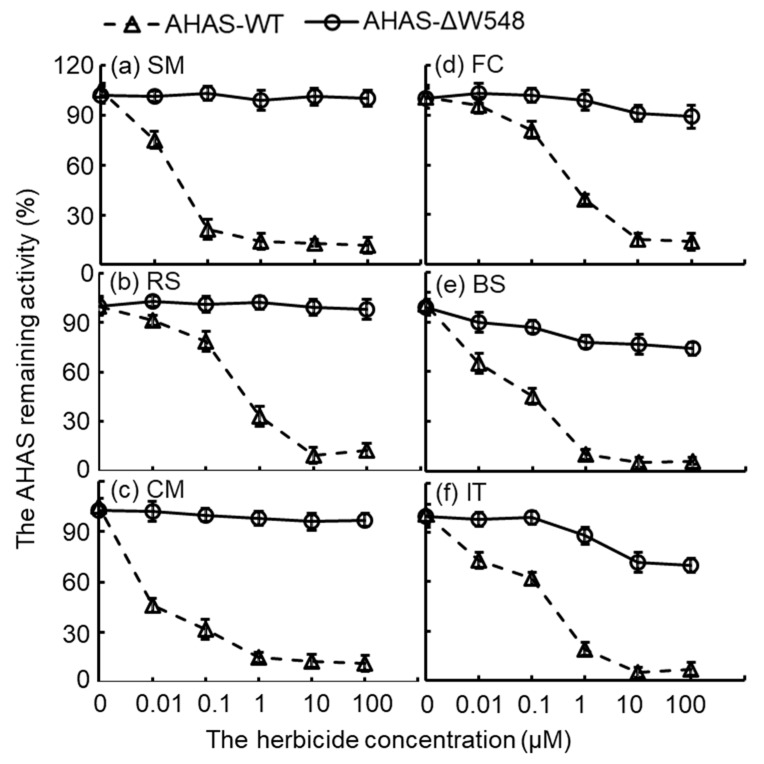
Bioassay curves of AHAS activities in the presence of six herbicides. AHAS-WT was inactive when an herbicide was more than 10 μM. AHAS-ΔW548 remained active at high concentrations of herbicides. Panels of (**a**–**f**) displayed the AHAS remaining activities in solutions of different herbicides: (**a**) sulfometuron-methyl (SM), (**b**) rimsulfuron (RS), (**c**) chlorimuron-methyl (CM), (**d**) flucarbazone-sodium (FC), (**e**) bispyribac-sodium (BS), and (**f**) imazethapyr (IT). Error bars represented the errors from triplicate measurements. Triangles are AHAS-WT, and circles are AHAS-ΔW548.

**Figure 5 ijms-21-01265-f005:**
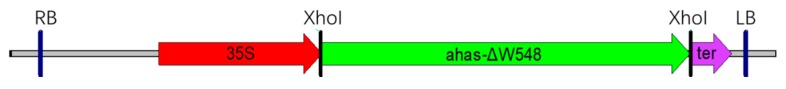
Diagram of T-DNA cassette for rice transformation. The *ahas-ΔW548* gene replaced the hygromycin resistant gene and served as a selection marker. This gene was driven by a 35S promoter. RB and LB: right and left border of T-DNA; 35S: 35S promoter of the CaMV; ter: terminator of the CaMV.

**Figure 6 ijms-21-01265-f006:**
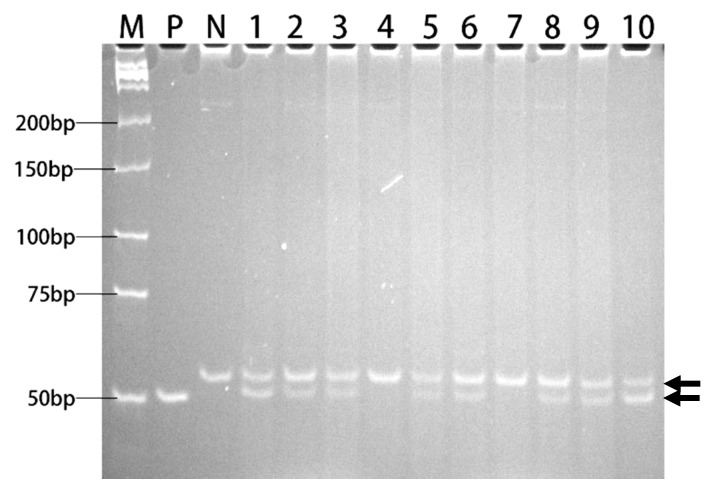
PAGE analysis of PCR products for *ahas* fragments. Lane 1-10: PCR products from Line 1-10 of transgenic rice. P: positive control (plasmid DNA of 1300-AHAS-ΔW548). N: negative control (genomic DNA of Xiushui134). M: Low MW DNA marker-A. The molecular weights were indicated. Two arrows showed two amplified fragments: upper arrows indicated wildtype *ahas* fragments, lower arrows indicated *ahas-ΔW548* fragments.

**Figure 7 ijms-21-01265-f007:**
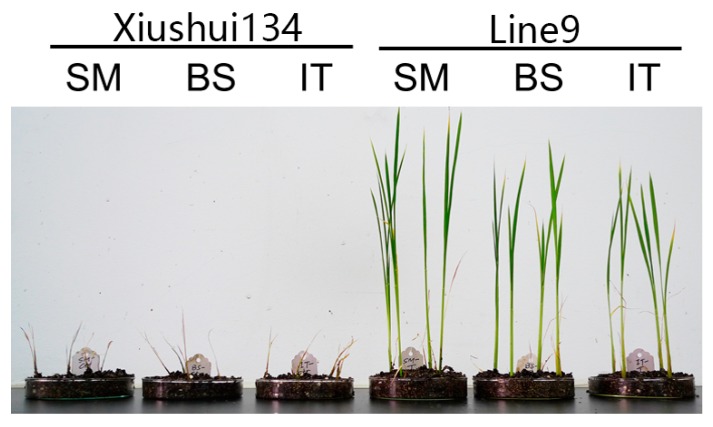
Image showing rice plants with herbicide treatment. The plants were treated with 0.25 g/L SM, 0.8 g/L BS or 2.2 g/L IT at the seedling stage. After two weeks, the Xiushui134 plants withered, while the Line9 plants survived with the herbicide treatment.

**Figure 8 ijms-21-01265-f008:**
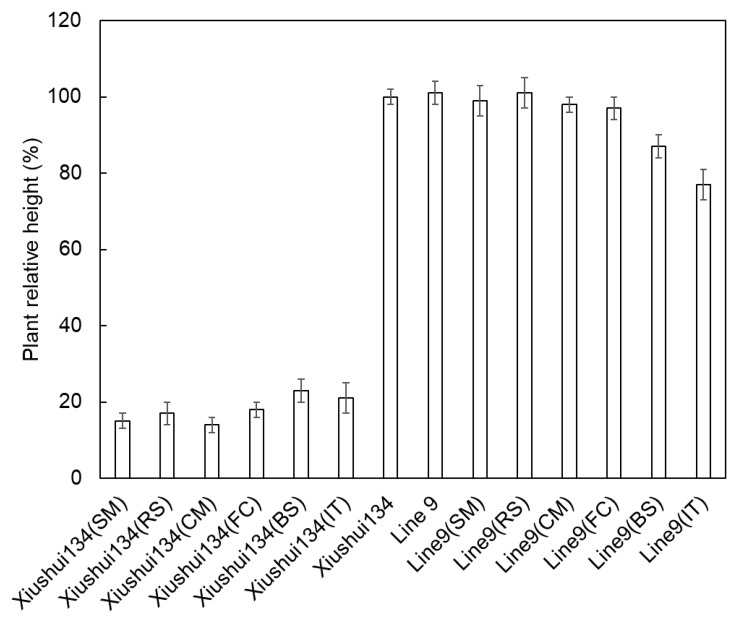
Plant relative height of Xiushui134 and Line9 at the seedling stage. The Xiushui134 plants were killed with herbicide treatments, and the relative heights were around 20%. The Line9 plants grew well with SU herbicide treatments. The growth of Line9 was inhibited by a PYB (BS) and IMI (IT) herbicides. Error bars represent the errors from multiple measurements.

**Figure 9 ijms-21-01265-f009:**
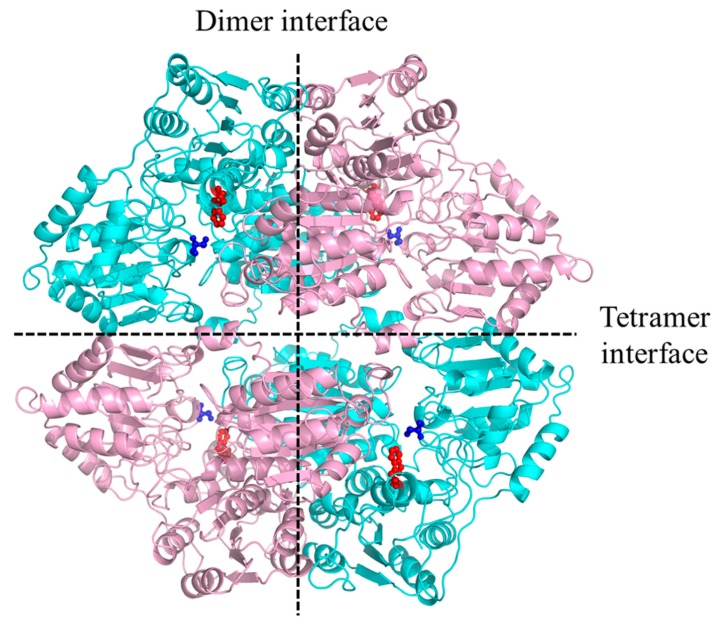
The W548 location in the AHAS-WT tetramer. The interfaces were labeled by dash Lines. The backbone of the monomer was colored as pink and cyan, respectively. W548 was shown as a red stick-ball model, S627 was shown as a blue stick-ball model.

**Table 1 ijms-21-01265-t001:** Molecular docking scores for herbicides in different AHAS.

Herbicides	AHAS-WT	AHAS-ΔW548
SM	7.159	5.731
RS	7.491	5.834
CM	7.507	5.784
FC	7.681	6.619
BS	7.208	6.389
IT	7.641	7.544

**Table 2 ijms-21-01265-t002:** Kinetic parameters of the AHAS-WT and AHAS-ΔW548 enzymes.

Enzyme/Herbicide ^a^	*K_m_* (mM) ^b^	*K_cat_* (s^−1^)	Catalytic Efficiency *K_cat_/K_m_* (S^−1^mM^−1^)
AHAS-WT	9.2 ± 0.4	6.7 ± 0.3	0.71 ± 0.03
AHAS-ΔW548	9.4 ± 0.4	6.6 ± 0.5	0.70 ± 0.05
AHAS-ΔW548 (SM)	9.3 ± 0.6	6.6 ± 0.3	0.72 ± 0.05
AHAS-ΔW548 (RS)	9.4 ± 0.5	6.5 ± 0.4	0.69 ± 0.04
AHAS-ΔW548 (CM)	9.2 ± 0.3	6.5 ± 0.4	0.71 ± 0.03
AHAS-ΔW548 (FC)	9.4 ± 0.3	5.9 ± 0.4 *	0.63 ± 0.03 *
AHAS-ΔW548 (BS)	11.3 ± 0.6 **	5.8 ± 0.3 **	0.51 ± 0.04 **
AHAS-ΔW548 (IT)	12.1 ± 0.3 **	5.1 ± 0.3 **	0.42 ± 0.03 **

Values are given as means ( ± standard deviation). * represented significantly different from AHAS-WT without an herbicide treatment at *p* < 0.05, ** represented significantly different from AHAS-WT without an herbicide treatment at *p* < 0.01. a: The enzyme was measured in the absence or presence of an herbicide. An herbicide was indicated in a bracket. b: ***K_m_*** means the Michaelis-Menten constant.

**Table 3 ijms-21-01265-t003:** Agronomic traits of the Xiushui134 and Line9 plants.

Line Name	Plant Height (cm)	Panicle Numbers Per Plant	Length of Panicle (cm)	Grains Per Panicle	Seed-Setting Rate (%)	1000-Grain Weight (g)	Yield Per Plant (g)	Grain Width(cm)	Grain Length(cm)
Xiushui134	94.1 ± 1.3	8.6 ± 0.4	14.2 ± 1.8	151.2 ± 9.6	90.2 ± 1.5	26.9 ± 0.7	28.5 ± 2.9	3.3 ± 0.2	6.8 ± 0.3
Line9	92.2 ± 2.1	8.9 ± 0.5	14.8 ± 1.6	145.9 ± 8.7	91.5 ± 1.3	27.2 ± 1.5	29.8 ± 3.6	3.3 ± 0.2	6.9 ± 0.5
Line9 (SM)	93.5 ± 1.8	8.5 ± 0.5	15.1 ± 1.2	153.5 ± 10.8	90.5 ± 2.1	26.3 ± 1.3	27.3 ± 2.1	3.3 ± 0.2	6.8 ± 0.4

Values are given as means (±standard deviation). The SM was applied to estimate the effects of an herbicide on the Line9.

**Table 4 ijms-21-01265-t004:** Total protein and free Amino Acids in the Xiushui134 and Line9 seeds.

Compound	Xiushui134	Line9	Line9 (SM)	Compound	Xiushui134	Line9	Line9 (SM)
Ile ^a^	18 ± 8	79 ± 26 **	75 ± 23 **	Trp	89 ± 29	95 ± 21	85 ± 31
Leu	23 ± 11	82 ± 25 **	87 ± 29 **	Gly	103 ± 20	105 ± 23	99 ± 31
Val	51 ± 17	224 ± 32 **	231 ± 28 **	Gln	145 ± 29	167 ± 35	154 ± 31
Phe	22 ± 9	25 ± 7	29 ± 6	Pro	154 ± 31	176 ± 30	181 ± 27
Lys	35 ± 11	28 ± 9	31 ± 12	Ala	266 ± 29	311 ± 35	304 ± 33
His	38 ± 16	47 ± 8	46 ± 9	Ser	270 ± 21	302 ± 29	298 ± 37
Thr	42 ± 9	45 ± 11	49 ± 13	Arg	340 ± 31	362 ± 49	388 ± 37
Met	46 ± 20	51 ± 12	41 ± 22	Asp	513 ± 46	501 ± 56	481 ± 43
His	48 ± 17	41 ± 21	59 ± 15	Asn	759 ± 57	780 ± 62	776 ± 69
Tyr	75 ± 36	89 ± 22	95 ± 32	Glu	1527 ± 246	1382 ± 321	1659 ± 356
Total Protein ^b^	8.9 ± 0.5	9.1 ± 0.7	9.2 ± 0.6				

Values are given as means (±standard deviation). The SM was applied to estimate the effects of an herbicide on the Line9. a: Unit of free amino acids was nM. b: Unit of total protein was %. ** represented significantly different from the Xiushui134 at *p* < 0.01.

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
