# Peer review of "Engineering Herbicide-Tolerance Rice Expressing an Acetohydroxyacid Synthase with a Single Amino Acid Deletion"

_ijms, 2020, doi:10.3390/ijms21041265_

Round 1

Reviewer 1 Report

The manuscript, Jun Fang  et. al ( I.D.: ijms-715169) describes a novel AHAS synthase enzyme, the expression of which in rice plants can contribute to multi-family herbicide tolerance.

I have some comments and suggestion for the authors.

Line 44: IMI: please specify and add a complete name

Line 65: Please check the reference 18 at line 397.

Line 76: Please add the gene number of the AHAS gene

Line 112: Please add SDS-PAGE, otherwise it is not clear that it is denaturing or non-denaturing gel.

Line 116: Please add arrows for the peaks to increase the visibility. Indicate the dimer peak (128kDa) and tetramer peak (256kDa)

Line 133: On Figure 4, panel d; please replace the title of FS to FC.

Line135: What do “replicate measurements” mean?  How many repetitions were performed? If you calculate the error bars, at least three measurements are necessary.

Line 137, Table 3. You have to define the “Km” at the first appearance in the text. I see that the authors did this at line 290, but please move the explanation to the first mentioning.

Line 146: Authors wrote in the line 145: “Eighteen lines survive” The reviewer question is: Why only one line (Line9) was tested by the authors? Single line is not enough to make any conclusion of the function of an integrated gene.

To test any type of gene functionality as a transgene expression, at least two or three independent transformed lines are necessary for testing, to avoid any other affect that is not related to the integrated transgene (e.g.: somaclonal variability, position effect variegation, chromosomal rearrangement, natural mutation…) Have the authors got information of the other seventeen lines? More details about these lines should be added as a supplement file at least.

Why did not the authors test the expression of the integrated transgene? This step is always necessary when any type of transgenic plants are created, tested (dose dependent pheotype)  etc...

Please specify, what “Xiushui134” is, or add reference regarding it.

Authors wrote: “One line, Line9, displayed similar agronomic traits as Xiushui134” How is this possible? Please explain it. Why does only one line show similar agronomic traits as wild type Xiushui134? What about the rest of the lines?

Line 159: Figure 6: Please use arrows, and notes in figure legend about which band represents which fragment, to increase visibility.

Line 163: Figure 7. Please add a short description of the treatment in figure legend

Line 169:  What do “replicate measurements” mean? How many plants were used for each experiment, and how many independent experiments were performed by the authors?

Line 170: Table 4: Same as before, in Line 169: How many plants were used for each experiment, and how many independent experiments were performed by the authors?

Line 173: Again as before:

How many plants were used for each experiment, and how many independent experiments were performed by the authors?

In the materials methods, at Line 328-329, the authors used rice seeds instead of plants for this experiment. Please correct this.

Line 213:  Authors wrote: “Some protein overexpression might produce negative agronomic traits.”  Please add a more detailed information about this problem. Please add some statistic and experimental data e.g: expression data of the integrated AHAS-ΔW548, etc.. about the rest of the lines. If you test eighteen lines, and only just one is promising, and the rest have a “negative” effect, please try to explain why it would be good to use this system (overexpression of AHAS- ΔW548) to generate herbicide resistant plants.

Line 215: Authors wrote: “that the AHAS-ΔW548 did not affect rice growth”. This statement is not true, please read my previous comment.

Line 253,300,301,309,310 : Please move all the primers used in this study into a single excel file.

Line 263: pGEXT2, please specify and add the name of the manufacturer company. In the Methods section, please check carefully, and add the name of the manufacturer for all chemicals, cells, etc. where this information is missing.

Line 274: Please add reference

Author Response

Dear Reviewer,

Please see the attachment. Thanks.

Sincerely,

Jun Fang

Reviewer 2 Report

Dear Authors,

I had a great opportunity to review research manuscript entitled: “Creation of herbicide-tolerance rice expressing an acetohydroxyacid synthase with a single amino acid deletion”, which is considered for publication in International Journal of Molecular Sciences. I analyzed whole manuscript which in my humble opinion presents some interesting insights in formation of new herbicide tolerant mutant of rice. However, level of manuscript preparation, results presentation for now is not enough for publication standards and publication rules of International Journal of Molecular Sciences. Therefore, I would like suggest major revision of manuscript. Reason for that decision I present in a form of list of specific comments.

General comment: I do not want to argue with authors but generation of crop plants (like rice or other plants) in a way presented in article is little strange. Because in general, herbicides (which has specific active compounds) use should destroy weed plants which lower production of crop plants. However weed plants quite quick develop resistance to active compounds in specific herbicides (because of selective pressure). So during cultivation farmer need to make a lot of rotation of types of herbicides with different type active compounds which has different biochemical effect on weeds. In authors research authors develop rice herbicide tolerant to different types of herbicides but with the same biochemical mechanism of function (blocking of amino acid synthesis) so use of this mutant rice plant has only short term practical usage because weeds develop resistance quick for this herbicides. Therefore use of this herbicides will be useless in future. And then work need be done form start all over again. Therefore, in the word research during creation of herbicide tolerant crop plants researchers develop tolerance for herbicides with different type of biochemical mechanisms to enable long term rotation of active compounds.

Title Section

Title is not formulated in scientific way. Term “creation” is rather biblical not strictly scientific. I would like suggest reformulation of the title in form for example: “Various type herbicide-tolerance rice with a single amino acid deletion anacetohydroxyacid synthase encoding gene. Or something else but without term “creation”.

Introduction section.

Authors must add information about most popular herbicides used during rice cultivation with active compounds and the biochemical effect on plant it is crucial for context of article.

Results section

One of the best part of manuscript but also need updates:

Table 1 must be removed or add appropriate citation. Reason of that is simple Table 1 did not presents author result but only dat form PubChem database

Figure 1: Is poor quality need to be larger. Moreover authors should present whole structure and then active bind sites

Discussion section

Is too short in comparison for amunt of results presented by authors. It need to be extended for dipper understand of authors results

Materials and methods section

This section is very problematic and prepared in completely not in clear way. Moreover, this section is not prepared according journal publication rules. Authors did not read journal rulesrules?? Why this section is named only methods. Why this section has only 4 citation. Authors develop all methods from the start I doubt it is possible. Authors must rewrite this cection in way that enables repeating of results. All bioinformatics programs must have appropriate citation etc. Authors did not mentioned about exact statistic analyses of results from Figure 4, Table 3, Figure 8, Table 4 and 5

Conclusion and references section.

Journal rules did not allow to add citations in conclusion section. Whats worse any of 43 position in reference list is not prepared according journal rules. References are prepared in sloppy way and scientific level and prestige of International Journal of Molecular Sciences did not allow sloppiness.

Sincerely,

Author Response

(The authors gave the same response as above.)

Reviewer 3 Report

In the course of described work new variety of herbicide tolerant rice was developed. The work is quite interesting but the paper needs some improvements before it could be published.

I would like to see some clarification how this is supposed to help with weedy rice tolerance to herbicides? Is weedy rice not tolerant to some of those herbicides?

How the developed variety is different from those previously generated?

The table legends are not sufficient. In particular this concerns Table 1. The description in the text doeas not match the table legends. In the text it is is stated that it contains list of  herbicides. In  table it is stated that we have homologous structures. There is no explanation where the numbers came from.  The manuscript should be written in such a way that it is easy to understand and reproduce the results. Some language correction is necessary as at times the meaning is not clear.

In the discussion there is too much repetition from the results section.

Author Response

(The authors gave the same response as above.)

Round 2

Reviewer 1 Report

The revised manuscript, Jun Fang  et. al ( I.D.: ijms-715169-peer-review-v2) describes a novel AHAS synthase enzyme, the expression of which in rice plants can contribute to multi-family herbicide tolerance.

The authors corrected all my previous suggestions. I have no more 
questions or suggestions.

Reviewer 2 Report

Dear Authors,

Authors adressed all my comments and improved manuscript. I recommend publication

Sincerely,